# The Effect of Micro-Focused Ultrasound Treatment on Intradermal Botulinum Toxin Type A Injection

**DOI:** 10.3390/toxins17030147

**Published:** 2025-03-19

**Authors:** Sarawin Harnchoowong, Natthachat Jurairattanaporn, Vasanop Vachiramon

**Affiliations:** 1Division of Dermatology, Department of Medicine, Faculty of Medicine, Ramathibodi Hospital, Mahidol University, 270 Rama VI Road, Ratchathewi, Bangkok 10400, Thailand; hsarawin@gmail.com (S.H.); natthachat.jur@mahidol.edu (N.J.); 2Department of Dermatology, Faculty of Medicine, Srinakharinwirot University, Bangkok 10110, Thailand

**Keywords:** hyperhidrosis, neuromodulator, neurotoxin, tightening, wrinkle

## Abstract

Botulinum toxin type A (BoNT-A) injection and micro-focused ultrasound with visualization (MFU-V) are becoming increasingly popular for various esthetic conditions. However, performing MFU-V subsequently to BoNT-A injection may diminish the efficacy of BoNT-A. Previous reports have indicated the impact of various energy-based devices on BoNT-A. Nevertheless, the influence of MFU-V on BoNT-A has yet to be investigated. Thus, this study aims to evaluate the effect of MFU-V on BoNT-A injection and to determine the optimal interval between each treatment. A randomized controlled study was conducted on 15 subjects. Each participant was assigned to receive four sites of intradermal BoNT-A injection on their forehead. Following BoNT-A injection, these locations were randomized to receive either no MFU-V treatment or further treatment with MFU-V (superficial transducer, 10.0 MHz, 1.5 mm) on the same day, Day 7, or Day 14. On Day 28 following injection, the anhidrosis area was measured using the iodine starch test to objectively evaluate the result. In comparison to the control site (1.10 cm^2^), the anhidrotic area was significantly smaller in all MFU-V treated sites (0.74 cm^2^ in immediate MFU-V treatment, *p* < 0.001; 0.90 cm^2^ in MFU-V Day 7, *p* = 0.005; 0.93 cm^2^ in MFU-V Day 14, *p* = 0.021). There were no serious and esthetic complications reported in our study. **In conclusion,** MFU-V with a superficial depth transducer significantly reduced the efficacy of intradermal BoNT-A injection, especially within two weeks following BoNT-A injection. After BoNT-A injection, we recommend MFU-V treatment should be avoided for at least two weeks. Further study is required to evaluate the optimal interval between these two treatments.

## 1. Introduction

Botulinum toxin type A (BoNT-A) has been commonly used and approved for a number of indications, including neuromuscular disorders and esthetic purposes. It is also effective and indicative in the treatment of hyperhidrosis. BoNT-A is a neurotoxin complex produced by *Clostridium botulinum* [1]. BoNT-A acts at cholinergic nerve terminals of neuromuscular junctions and autonomic innervation of the sweat glands. It splits soluble N-ethylmaleimide-sensitive factor attachment protein receptors (SNAREs), which blocks the docking of neurotransmitter-containing vesicles to the synaptic membrane and blocks the release of acytylcholine [1]. BoNT-A also reduces responsiveness of the sweat glands to acetylcholine, which leads to its efficacy in hyperhidrosis treatment [2,3]. IncobotulinumtoxinA (IncoBoNT-A) is one available BoNT-A on the market. IncoBoNT-A is highly purified with smaller molecules. It contains only 150 kD neurotoxin without complexing proteins. Hence, injection with IncoBoNT-A reduces the chance of developing neurotoxin antibodies, which can result in toxin resistance [3,4].

Micro-focused ultrasound with visualization (MFU-V) is a widely used energy-based device for skin tightening. It delivers focused energy pulses to the deep reticular dermis and superficial musculoaponeurotic system without damaging overlying epidermis. This energy heats the targeted tissue temperature up to greater than 60 °C and produces thermal coagulation. As a result, MFU-V contracts and promotes the synthesis and remodeling of a new collagen [5].

To achieve the best result in skin rejuvenation, each patient could require a combination of treatment modalities. BoNT-A is composed of heat-labile disulfide bonds [6]. Thus, there is a hypothesis that combining energy-based devices with BoNT-A results in an inactivation of BoNT-A. Previous studies reported the effect of several energy-based devices on BoNT-A injection [7]. There is no study conducted focusing mainly on the effect of MFU-V on BoNT-A injection. Thus, in this study, we aim to investigate the effect of MFU-V on IncoBoNT-A injection. We utilize objective parameter, the anhidrosis area, to thoroughly demonstrate this effect. To assess the appropriate interval between both treatments, different time intervals were examined in our study.

## 2. Results

A total of 15 participants were eligible and enrolled. All subjects’ demographic data were provided in Table 1. A total of 14 out of 15 participants (93.3%) were female. Mean age was 33.5 ± 9.4 years. Two thirds of participants (66.7%) were classified as Fitzpatrick skin type III while 33.3% were classified as Fitzpatrick skin type IV. The majority of participants (73.3%) received BoNT-A injection for the first time. Others had received BoNT-A injection for more than 6 months and had a homogeneous iodine starch test before the start of the trial. None of the patients had previously been treated with focused ultrasound.

For the primary outcome, a summary of the anhidrosis area is shown in Table 2. The anhidrosis area produced by the botulinum toxin was 1.10 ± 0.33 cm^2^ for site A, the control site without MFU-V treatment. Site B, which received MFU-V treatment immediately after BoNT-A injection, had the smallest anhidrosis area, measuring 0.73 ± 0.35 cm^2^, followed by 0.89 ± 0.32 cm^2^ at site C (MFU-V treatment seven days after toxin injection) and 0.93 ± 0.30 cm^2^ at site D (MFU-V treatment 14 days after toxin injection). Compared to the control site, all MFU-V treated areas, site B, C, and D, showed significantly smaller anhidrosis size with a *p*-value of <0.001, 0.005, and 0.021, respectively. In comparison between MFU-V treated group, the immediate treatment site had a statistically significant smaller anhidrosis area, compared to sites with subsequent MFU-V treatment on Day 7 and 14 (*p*-value 0.040, 0.022, respectively), as presented in Figure 1.

Most of the participants experienced minor adverse events such as discomfort, erythema, and edema, which typically occurred after both treatments. All cases of erythema and edema were resolved in a few days without any complication. Eyelid ptosis, asymmetry in frontal lines, or other serious complication were not observed in our study.

## 3. Discussion

To fulfill successful facial rejuvenation, a combination of multiple modalities is needed. Generally, treatment with injectables including BoNT-A and fillers were recommended as the final step in the rejuvenation procedure. This is due to the fact that performing fillers or BoNT-A with a laser at the same time may increase the possibility of filler–laser or toxin–laser interactions and further complications. BoNT-A is the heat-labile substance [6]. As a result, the heat generated by any energy-based devices may inactivate or destroy the toxin, therefore, attenuating its effects. Furthermore, energy-based devices could manipulate the toxin to migrate toward the undesirable target, resulting in complication and facial asymmetry [8,9]. However, there were several studies reporting additional benefits from BoNT-A in combination with laser, intense pulsed light, and radiofrequency devices. Prior studies reported that adjuvant BoNT-A treatment sites improved wrinkles and rhytides more effectively than energy-based device treatment alone, with no additional complications [10,11]. In contrast, studies on the effect of energy-based devices on toxin injection are more limited. To the best of our knowledge, our study is the first study to evaluate the effect of MFU-V effect on intradermal BoNT-A injection. For adequate assessment on this effect, we performed iodine starch test to measure the anhidrosis area produced by BoNT-A injection as an objective outcome.

We found that if MFU-V treatment was performed at the BoNT-A injection site within 14 days, it significantly diminished the effect of intradermal BoNT-A injection. In comparison to the control site, our study showed significantly reduced anhidrosis area in all MFU-V treated sites. The possible mechanisms can be explained by the effect of MFU-V and BoNT-A injection on targeted tissues. Normally, after injection, BoNT-A molecules bind quickly to the receptors in the local injection site and diffuse slowly to the surrounding area. Within 2 weeks of the toxin injection, the anhidrosis area steadily increased due to gradual diffusion. The anhidrosis area increased exponentially during the first 7 days. Then, it slowly increased until 14 days after injection [12]. MFU-V treatment during this period may inactivate portions of BoNT-A or inhibit the diffusion of BoNT-A by the heat from MFU-V. To achieve its effect and outcome, the ultrasound treatment heats the targeted tissues to 60–70 °C. The botulinum toxin in foods was demonstrated to be inactivated by heating up to 85 °C for 5 min. This explained why the anhidrosis region was noticeably smaller in all MFU-V treated sites, particularly the immediate treatment site [13,14]. Secondly, apart from collagen contracture, there are thermal coagulation areas created by MFU-V in different levels, depending on transducer depth [15]. Subsequently, the thermal dermal injury stimulates inflammatory and wound-healing responses. The significant inflammatory response at the thermal injured site occurred as early as 2 days to 10 weeks after treatment. During this inflammatory phase, macrophages play an important role in phagocytosis and cytokines production. This pronounced inflammation may associate to an inactivation or destruction of the toxin, illustrating the result shown in our study [5,14,16,17,18]. However, Nestor et al. reported the opposite result. The study by Nestor et al. demonstrated the effect of MFU-V on eccrine glands. The treatment of axillary hyperhidrosis with MFU-V with 3 mm and 4.5 mm transducers was effective and produced long-lasting results [19]. The explanation might be due to the different depth of transducers between the two studies.

Though there is currently no research on how MFU-V affects intradermal BoNT-A injection, there are a few studies about the effect of other energy-based devices on toxin injection. Semchyshyn et al. reported on 19 subjects who underwent toxin injections on glabellar or crow’s feet areas. One side was treated with either pulsed dye laser, intense pulsed light, 1450 nm diode laser, long-pulsed Nd:YAG, or a radiofrequency device, within 10 min after injection. The outcome demonstrated that BoNT-A was unaffected by these devices. Nevertheless, this study only used photographic comparison, which may not fully address the effect of energy-based devices on toxin injection [7]. The other study by Paul et al. also demonstrated that the 800 nm diode laser had no effect on BoNT-A injection. They used gravimetry to compare axillary sweating in nine patients with axillary hyperhidrosis. One site was treated with a 800 nm diode laser for hair reduction 1 week before BoNT-A injection, while the other site was treated immediately after toxin injection [20]. However, on the other hand, there were various reports of utilizing botulinum toxin injection and energy-based devices on the same day. Most of the studies demonstrated benefits of combining two treatments without reports of the effect of energy-based devices on toxin injection [10]. Recently, Jiang L et al. reported the effect of microneedle fractional radiofrequency on toxin injection. The authors demonstrated reduced efficacy of BoNT-A on crow’s feet in group immediately treated with BoNT-A injection following microneedle fractional radiofrequency treatment. The study group further worked on the mouse model and discovered that a 3-day interval is optimal for BoNT-A injection following microneedle radiofrequency treatment [21]. Apart from energy-based devices, there was one study by Sycha et al. reporting the effect of UV-B irradiation on intradermal BoNT-A injection. The study included six participants with intradermal BoNT-A injection on their thighs. Later, one site was irradiated with UV-B (using 3-time MED energy). The effect was evaluated on the anhidrosis area with an iodine starch test. At 2 weeks after injection, the result indicated a 30 percent reduction in the cutaneous effects of BoNT-A. This was explained by photodegradation and inactivation of BoNT-A by UV-B, the indirect proteotoxic effect of UV-B through reactive oxygen species and strong inflammation following UV-B irradiation [22].

Our study has some limitations. First, as pilot study design, we recruited quite a small number of participants. Second, anhidrosis area on different areas of forehead may be different in sizes [23]. Thus, we randomized all four sites to received different assigned treatments to minimize this bias. Lastly, the intervals between both treatments were limited to 14 days. Further study is needed to evaluate the appropriate interval of MFU-V treatment following toxin injection.

## 4. Conclusions

In conclusion, our study reported that MFU-V treatment with a superficial depth transducer attenuates the effect of intradermal BoNT-A injection on forehead sweating, especially if MFU-V treatment was performed within 14 days after toxin injection. We recommend that MFU-V treatment should be avoided for at least 14 days after BoNT-A injection.

## 5. Materials and Methods

### 5.1. Study Design

A prospective, randomized controlled study was conducted at Ramathibodi hospital, Thailand from May 2021 to May 2022. The study received the institutional review board approval from the Committee of Human Rights Related to Research Involving Human Subjects, Ramathibodi Hospital, Mahidol University (Protocol number MURA2021/398) and Thai Clinical Trial Registration number TCTR20211029004). All participants were informed about the study protocols, possible outcomes, and adverse events before the study enrollment. Then, the written informed consent was obtained.

### 5.2. Patients

As a pilot study, 15 participants were enrolled in the study. We included subjects above the age of 18-year-old with homogeneous sweating on the forehead, which was tested with the iodine starch test. Participants with concurrent skin inflammation or infections, scars, underlying immunocompromised or neuromuscular conditions, a history of forehead surgery within 6 months, history of forehead BoNT-A injection within 6 months, history of forehead filler injections within 12 months, and concurrent use of drugs effecting sweating and pregnancy or lactation were excluded.

### 5.3. Randomization, Allocation, and Concealment

A randomized list for treatment protocol was generated prior to the start of the research using block randomization from the website www.sealedenvelope.com (accessed on 31 January 2021). The randomized list was kept by the dermatologist who was not involved in patients’ assessment. The same dermatologist performed IncoBoNT-A injection and MFU-V treatment as assigned in the randomized list. Individual participants received four injections on their forehead. Then, all four injection points were randomized to receive different MFU-V treatment protocols as a randomized list in order from the right side of participants’ foreheads to the left side. Site A was the control, receiving no further treatment. Site B, C, and D were treated with MFU-V immediately (within 10 min after injection), 7 days, and 14 days after toxin injection, respectively. Figure 1 shows the sample of assigned injection sites on each participants’ forehead. The other dermatologist, who was blinded to the randomized list and patient’s treatment protocol, performed an iodine starch test and took a photograph before the treatment and on the final evaluation day. A protocol flowchart of the study is shown in Figure 2.

### 5.4. Treatment Procedure

Prior to enrollment, the iodine starch test was used to screen each subject. Individuals with homogeneous tests were then allocated to receive BoNT-A injection. The forehead of each subject was divided into four sites. The reference points were 2 cm above the eyebrows and 1.5 cm laterally to reference lines vertical to mid-pupils. However, in order to prevent complications, injection sites may be modified based on individual anatomy. All four sites were assigned to receive two units of intradermal IncoBoNT-A injection (Xeomin^®^, Merz Aesthetics, Raleigh, NC, USA, 1:2.5 concentration, dilution of 100 units of IncoBTX-A with 2.5 mL of normal saline). The injection was performed by one experienced dermatologist, in order to minimizing operator bias. IncoBoNT-A was prepared by sterile dilution in preservative-free saline (0.9% NaCl) and injected with hubless 31-guage needle syringes. Then, MFU-V was performed in accordance with randomization. The following treatments on Day 7 and Day 14 were performed by the same dermatologist. A 1.5 mm depth MFU-V transducer (Ultherapy, Merz Aesthetics, Raleigh, NC, USA) was used. The parameter was as follows: energy of 0.18 J, frequency of 10 Hz, treatment length of 1 cm, total of 15 lines (2 passes).

### 5.5. Outcome Assessment

The anhidrosis area was used to evaluate the primary outcome. It was measured with an iodine starch test at the baseline and 28 days after the injection. After the forehead was cleaned and dried, an iodine starch test was conducted. After applying 3% Lugol’s solution to the entire forehead and allowing it to dry, corn starch was administered. Sweating was induced by keeping participants in a temperature- and humidity-controlled room until the iodine starch test turned a homogenous blue-black coloration, indicating sweating. Lugol’s solution retained its yellow-brown hue in the anhidrosis area. After performing the iodine starch test on forehead, the anhidrosis area was captured by VISIA^®^ (Canfield Scientific, Morris County, NJ, USA) and a digital camera (Canon EOS 80D, Canon Inc., Tokyo, Japan). Then, the anhidrosis area was measured by ImageJ^®^ Version 1.53m (National Institutes of Health, Bethesda, MD, USA).

### 5.6. Adverse Effects

Adverse events were recorded on all procedural visits. Every participant was monitored for pain and other possible adverse reactions throughout the study. At the baseline and the end of the study (28 days after injection), a photographic assessment of forehead lines was also evaluated using a digital camera (Canon EOS 80D, Japan).

### 5.7. Statistical Analysis

Categorical data were presented as frequency (percentage). In contrast, numerical continuous data, such as measured anhidrosis area, were reported as mean ± standard deviation (for normal distribution data) and median with interquartile range. STATA/SE version 14.2 (STATA Corp., College Station, TX, USA) was used for data statistical analysis. Multilevel mixed effect ordered logistic regression was used to analyze the difference in anhidrosis area between each site. Statistical significance was defined as a *p*-value less than 0.05.

## Figures and Tables

**Figure 1 toxins-17-00147-f001:**
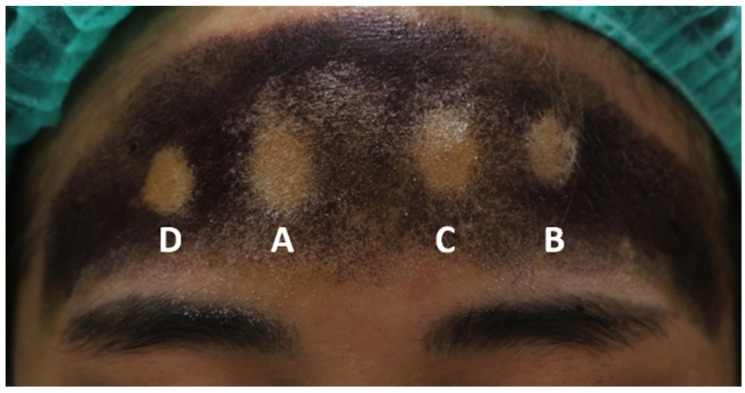
Photograph of anhidrosis area, A, control site, no MFU-V treatment; B, immediate MFU-V treatment; C, MFU-V treatment 7 days after BoNT-A injection; D, MFU-V treatment 14 days after BoNT-A injection, MFU-V, micro-focused ultrasound with visualization; BoNT-A, botulinum toxin A.

**Figure 2 toxins-17-00147-f002:**
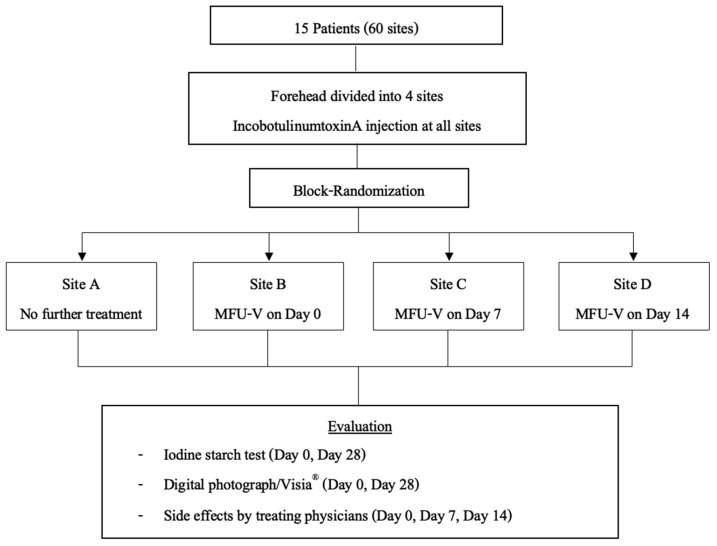
The flowchart of the study protocol; MFU-V, micro-focused ultrasound with visualization.

**Table 1 toxins-17-00147-t001:** Baseline subject characteristics.

Data	Value
Mean age, year (±SD)	33.5 (±9.4)
Sex, *n* (%)	
Male	1 (6.7%)
Female	14 (93.3%)
Skin type, *n* (%)	
III	10 (66.7%)
IV	5 (33.3%)
History of BoNT-A injection, *n* (%)	
Yes	4 (26.7%)
No	11 (73.3%)
History of focused ultrasound treatment, *n* (%)	
Yes	0 (0%)
No	15 (100%)

BoNT-A, botulinum toxin A; SD, standard deviation.

**Table 2 toxins-17-00147-t002:** Data comparing anhidrosis area between treatment sites.

**Site**	**Anhidrosis Area (cm^2^) (±SD)**	***p*-Value**
A (No MFU-V treatment)	1.10 (±0.33)	
B (Immediate MFU-V treatment)	0.73 (±0.35)	<0.001 *
C (MFU-V treatment 7 days after BoNT-A injection)	0.89 (±0.32)	0.005 * 0.040 ^†^
D (MFU-V treatment 14 days after BoNT-A injection)	0.93 (±0.30)	0.021 * 0.022 ^†^

MFU-V, micro-focused ultrasound with visualization; BoNT-A, botulinum toxin A; SD, standard deviation * Significant difference from control site A; ^†^ significant difference from immediate MFU-V treatment site B.

## Data Availability

The original contributions presented in this study are included in the article. Further inquiries can be directed to the corresponding author.

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
