# Peer review of "The Effect of Micro-Focused Ultrasound Treatment on Intradermal Botulinum Toxin Type A Injection"

_toxins, 2025, doi:10.3390/toxins17030147_

Round 1

Reviewer 1 Report

Comments and Suggestions for Authors

The manuscript under review is highly intriguing, and I would like to encourage its potential publication in The Journal. Below are some suggestions for improvement:

1. Figure 1 and Table 2 Correlation: Figure 1 does not fully represent the results depicted in Table 2 (specifically, Area D does not appear significantly larger than Area B). Could you provide a photograph of another patient that demonstrates more definitive results?

2. Additional Discussion on Focused Ultrasound: Focused ultrasound has been shown to effectively destroy eccrine glands, offering treatment for localized hyperhidrosis (J Clin Aesthet Dermatol. 2014;7(4):14–21). Although the current manuscript presents a negative impact of focused ultrasound on the efficacy of botulinum toxin in treating hyperhidrosis, a brief discussion on the contrasting perspective would enhance the manuscript.

Author Response

Plesse see the attachment.

Reviewer 2 Report

Comments and Suggestions for Authors

well written.

Not only Xeomin but any other botulinum toxin might show same result

Author Response

Comment1: Not only xeomin but any other botulinum toxin might show same result.

Response: One of the limitation of this study is that we did not examined the effect of MFU-V and other different BoNT-A formulation. We have stated the limitation of this issue in the discussion section of the manuscript.

Reviewer 3 Report

Comments and Suggestions for Authors

toxins-3511976

Abstract

Line 14         The exact sites/areas of injection should be mentioned here

Introduction

Line 34                    Citation needed

Lines-35 34             BoNT/A is produced as a toxin complex by the bacteria, not as purified neurotoxin.  Also, a citation is needed here.

Line 37                    BoNT/A blocks the docking of vesicles, not inhibits

Line 40                    IncobotulinumtoxinA is written as all one word, no spaces

Lines 43-44             There is very little evidence that use of INCO leads to less antibody production (resistance).

Line 52                    Citation needed

Lines 55-56             The authors should cite and include discussion on this work:

Park, J. Y., Lee, J. S., Lee, S. R., & Lee, D. H. (2023). Combined Treatment with Micro-Focused Ultrasound with Visualization and Intradermal Incobotulinumtoxin-A for Enlarged Facial Pores: A Retrospective Study in Asians. Clin Cosmet Investig Dermatol, 16, 1249-1255. https://doi.org/10.2147/CCID.S402001

Materials and Methods

Was the study registered with any clinical trials database?  This should have been carried out.

Line 189                  The full web address should be given

Line 195 et seq                  A picture of the injection and treatment sites, with labels, would be useful here.  Figure 1 could be used and introduced at this point.

Line 207                  What does “1:2.5 Concentration” mean?

Line 220                  How was sweating induced?  Were controlled conditions used?  These should be explained in this section.

Results

Line 63                    These data should align exactly with those in Table 1.  The data in the table have been rounded.

The Legends for Tables 1 and 2 should be improved

Lines 90-93             More information should be provided on the adverse events.  These could be presented as a table

Discussion

Line 99                    Citation needed

Lines 107-108 and 132-133                    Please see earlier comment on the additional citation by Park et al. 2023

Lines 115-116                    Toxin diffuses after injection and then binds to the receptors.  Diffusion cannot occur if the toxin molecule is bound to a receptor.

Line 119                  Citation 10 is probably not the best here.  I recommend using the work of Hexsel, for example:

Hexsel, D., Dal, T., Dini, F., Zechmeister-do-Prado, D., & Hexsel, C. (2005). Diffusion, dispersion or action halos of botulinum toxins? A pilot study comparing two commercial preparations of type A botulinum toxins. J Am Acad Dermatol, 53(3 (Suppl. 2)), AB2.

Hexsel, D., Dal'Forno, T., Hexsel, C., Do Prado, D. Z., & Lima, M. M. (2008). A randomized pilot study comparing the action halos of two commercial preparations of botulinum toxin type A. Dermatol Surg, 34(1), 52-59. https://doi.org/10.1111/j.1524-4725.2007.34008.x

Hexsel, D., Brum, C., do Prado, D. Z., Soirefmann, M., Rotta, F. T., Dal'Forno, T., & Rodrigues, T. C. (2012). Field effect of two commercial preparations of botulinum toxin type A: a prospective, double-blind, randomized clinical trial. J Am Acad Dermatol, 67(2), 226-232. https://doi.org/10.1016/j.jaad.2011.08.011

Hexsel, D., Hexsel, C., Siega, C., Schilling-Souza, J., Rotta, F. T., & Rodrigues, T. C. (2013). Fields of effects of 2 commercial preparations of botulinum toxin type A at equal labeled unit doses: a double-blind randomized trial. JAMA Dermatol, 149(12), 1386-1391. https://doi.org/10.1001/jamadermatol.2013.6440

Lines 124-126                    Citation needed

There are studies (approx. 50) on the use of, for example, lasers with toxin that go back over 20 years:

Zimbler, M. S., Holds, J. B., Kokoska, M. S., Glaser, D. A., Prendiville, S., Hollenbeak, C. S., & Thomas, J. R. (2001). Effect of botulinum toxin pretreatment on laser resurfacing results: a prospective, randomized, blinded trial. Arch Facial Plast Surg, 3(3), 165-169. https://doi.org/10.1001/archfaci.3.3.165

The authors should consider the whole literature available in their discussion.

Comments on the Quality of English Language

Can be improved.  The correct tense should be used.

Round 2

Reviewer 3 Report

Comments and Suggestions for Authors

None.  Authors have improved the manuscript based on my recommendations.